# Changes in Sexual Functioning in Women with Severe Obesity After Bariatric Surgery: Impact of Postoperative Adherence to Mediterranean Diet

**DOI:** 10.3390/nu17040605

**Published:** 2025-02-07

**Authors:** Jaime Ruiz-Tovar, Gilberto Gonzalez, Maria-de-Lourdes Bolaños, Eva-María Lopez-Torre, Maria-Encarnación Fernández-Contreras, Jesús Muñoz, Carolina Llavero

**Affiliations:** 1San Juan de Dios Foundation, 28036 Madrid, Spain; elopezt@comillas.edu (E.-M.L.-T.); jmunozm@comillas.edu (J.M.); 2Health Sciences Department, San Juan de Dios School of Nursing and Physical Therapy, Comillas Pontifical University, 28036 Madrid, Spain; 3Hospital Real San José, Guadalajara 19001, Mexico; gilpchmd@yahoo.com.mx; 4Neuroscience Institute, Centro Universitario de Ciencias Biológico Agropecuarias (CUCBA), University of Guadalajara, Guadalajara 44600, Mexico; mariad.bolanosm@academicos.udg.mx; 5Health Sciences Department, Alfonso X University, 28691 Madrid, Spain; marifeco@externos.uax.es; 6Day Hospital Unit, Hospital Universitario del Henares, 28822 Madrid, Spain; carolinallavero@gmail.com

**Keywords:** severe obesity, bariatric surgery, female sexual function index, Mediterranean diet, adherence, lubrication, orgasm, satisfaction, type 2 diabetes mellitus, metabolic syndrome

## Abstract

Background: This study analyzes the effects of bariatric surgery on female sexual function, assessed using the Female Sexual Function Index (FSFI), and explores the impact of adherence to the Mediterranean diet during the postoperative period. Patients and methods: A retrospective observational study was conducted using a prospectively collected database, including heterosexual women with morbid obesity undergoing bariatric procedures. The FSFI questionnaire was applied before the intervention and 24 months after surgery. Adherence to the Mediterranean diet was evaluated using the PREDIMED questionnaire. Results: Among the 240 participants, 70.8% presented preoperative sexual dysfunction, which decreased to 20.5% two years post-surgery. Significant improvements were observed in all FSFI domains except for pain. Good adherence to the Mediterranean diet was associated with higher scores in the lubrication, orgasm, and satisfaction domains. Conclusions: Bariatric surgery significantly improves female sexual function, with the Mediterranean diet enhancing these benefits during the postoperative period. Future studies must investigate additional variables such as psychological factors, physical activity, and other lifestyle changes that may also influence sexual function.

## 1. Introduction

Obesity is associated with the onset of numerous physical and psychological comorbidities. Among the psychological impacts, depression, reduced self-esteem, poor quality of life, and heightened body dissatisfaction are particularly prevalent [1,2,3,4,5]. One of the most significant elements of quality of life is sexual functioning, yet it is frequently neglected due to the discomfort associated with discussing such matters. In individuals with morbid obesity, sexual dysfunction arises not only from psychological issues but also from additional pelvic floor complications, including urinary incontinence, pelvic organ prolapse, and fecal incontinence [6,7].

Sexual function can be adversely affected by obesity in both sexes; however, women with obesity often experience more significant and complex difficulties in this domain [8,9,10].

Self-administered questionnaires are frequently employed to evaluate sexual dysfunction, with the Female Sexual Function Index (FSFI) being among the most widely utilized tools. This comprehensive 19-item instrument assesses sexual function in women across six key dimensions: desire, arousal, lubrication, orgasm, satisfaction, and pain [11].

While numerous studies have noted enhancements in female sexual function associated with weight loss after bariatric surgery [12,13], such improvements are not consistently mirrored in substantial changes to FSFI scores [14]. A meta-analysis by Gao et al. [15] examined changes in sexual dysfunction following bariatric surgery using self-reported questionnaires, including the FSFI. The review identified only 16 studies reporting FSFI outcomes across 881 participants, with the largest sample comprising 106 individuals [16]. This underscores a notable limitation in current research, as numerous studies may not include sufficiently large sample sizes to identify significant enhancements in FSFI dimensions. On another note, robust adherence to the Mediterranean diet is widely recognized for its positive effects on both physical and mental health, contributing to greater longevity and a lower prevalence of obesity and cardiovascular conditions [17,18,19]. Research comparing different dietary patterns has identified the Mediterranean diet as the most effective model for reducing cardiovascular risk [20]. The Mediterranean diet is distinguished by its emphasis on consuming ample amounts of fruits, vegetables, legumes, nuts, and whole grains, alongside moderate fish intake and minimal consumption of meat and sweets. Olive oil serves as the principal source of dietary fats [21]. These dietary components are abundant in phytonutrients, such as polyphenols and essential vitamins. The Mediterranean diet also provides a rich source of antioxidants, including vitamin E, β-carotene, vitamin C, and flavonoids, alongside key minerals like selenium and folate in their natural forms [22]. Moreover, the Mediterranean diet is acknowledged for its environmental sustainability, making it a favorable choice from both health and ecological perspectives [23].

Severe obesity is frequently associated with metabolic syndrome and its related conditions, including type 2 diabetes mellitus, arterial hypertension, and dyslipidemia. The link between sexual dysfunction and metabolic syndrome is well established, with both significantly diminishing quality of life. Recent findings indicate that adherence to the Mediterranean diet among patients with metabolic syndrome can partially alleviate several forms of sexual dysfunction in women, such as those linked to menstrual irregularities, menopause, endometriosis, and polycystic ovary syndrome [24]. Giugliano et al. further observed that among women with type 2 diabetes, higher adherence to the Mediterranean diet correlated with a reduced incidence of female sexual dysfunction [25]. To the best of our knowledge, no studies to date have examined improvement in sexual function after bariatric surgery in relation to adherence to the Mediterranean diet.

This research aims to assess the influence of bariatric surgery on female sexual function, using the FSFI as a measurement tool, in what represents the largest case series of its kind to date. Furthermore, it investigates how adherence to the Mediterranean diet post-surgery impacts female sexual function.

## 2. Materials and Methods

This retrospective observational study utilized a database compiled prospectively. The cohort included heterosexual Spanish women who underwent Roux-en-Y gastric bypass (RYGB) or sleeve gastrectomy (SG) between September 2020 and September 2022 at a Spanish private institution. Eligibility criteria required participants to have a body mass index (BMI) > 40 kg/m^2^ or >35 kg/m^2^ with obesity-related comorbidities. Patients were excluded if they had undergone alternative surgical procedures, required revision surgeries, or had not provided informed consent for participation. Additional exclusion criteria encompassed nonambulatory individuals and those with a history of psychiatric disorders. Participants were required to be in a relationship lasting at least 12 months to ensure the opportunity for sexual activity with a partner. Cases lost to follow-up were omitted from the analysis. Furthermore, individuals without a partner or who had not engaged in sexual activity during the month preceding questionnaire completion were also excluded. A visual representation of the study design is shown in Figure 1.

### 2.1. FSFI Questionnaire

At the Outpatient Clinic, patients were requested to complete the Female Sexual Function Index (FSFI) questionnaire [11] at two time points: prior to the intervention and 24 months post-surgery. Assessing sexual function formed an integral component of the holistic patient evaluation conducted by the multidisciplinary team at our institution’s bariatric surgery unit, with a specialist nurse overseeing the distribution and analysis of the questionnaires.

The FSFI, comprising 19 items, assesses six distinct domains of sexual function: desire (range: 1.2–6); arousal (range: 0–6); lubrication (range: 0–6); orgasm (range: 0–6); satisfaction (range: 0.8–6); and pain (range: 0–6). The desire domain evaluates frequency and the level of interest through two specific questions. Arousal is measured via four items assessing frequency, intensity, confidence, and satisfaction. Lubrication is examined through four questions addressing frequency, difficulty, and the ability to sustain lubrication. The orgasm domain includes three items concerning frequency, difficulty, and satisfaction. Satisfaction is gauged through three questions focusing on closeness with a partner, the quality of sexual interactions, and overall satisfaction with sex life. Lastly, the pain domain evaluates the frequency and intensity of pain during and after vaginal penetration through three items. Scores for each domain are aggregated to yield a total score ranging from 2 to 36, with higher scores indicating better sexual function. A score of 23 or below denotes sexual dysfunction [11].

### 2.2. PREDIMED Questionnaire

Adherence to the Mediterranean diet was evaluated using the PREDIMED questionnaire [26], a tool specifically validated for Spanish-speaking populations. This questionnaire comprises a set of questions regarding dietary habits and food consumption patterns, with each response assigned a corresponding score. Patients were categorized into low-, medium-, or high-adherence groups based on their scores.

Patients completed the questionnaire both prior to surgery and 24 months after the procedure.

Throughout the postoperative period, patients received guidance from a specialized nutrition nurse regarding dietary progression. Initially, they followed a liquid diet for the first two weeks post-surgery, which was then gradually advanced to a semi-solid diet and eventually to a full solid diet by approximately three months post-surgery. The solid diet adhered to the Mediterranean diet framework, with the following characteristics: using olive oil as the main source of fat, consuming plenty of fruit, vegetables, pulses, and nuts, including wholemeal bread, pasta, rice, and other whole grain cereals in the daily diet, opting for low-processed, fresh, and locally sourced foods, consuming dairy products daily, such as low-fat yogurt and fresh cheese, consuming red meat in moderation, preferably as part of stews, eating plenty of fish and eggs in moderation, consuming fresh fruit as the usual dessert, and consuming water as the only beverage, although water should be drunk between meals, not during. Patients were instructed to eat slowly, chew food well, and divide meals into several small meals a day.

During the first year after surgery, patients attended quarterly consultations with the nutrition nurse to monitor adherence to the diet and address any issues with food tolerance. During the first year after surgery, patients usually have difficulties in eating red meat, rice, and pasta, which usually disappear 12 months after surgery, when they can usually eat all types of food without problems. After the first year, nutritional evaluations were conducted on an annual basis.

### 2.3. Additional Variables

Additional variables analyzed included age, marital status, surgical technique, and excess BMI loss 24 months after surgery.

### 2.4. Statistical Methodology

Quantitative variables were described using the mean and standard deviation, while qualitative variables were represented by the number of cases and percentages. For the analytical study of the variables, the Student’s *t*-test (Wilcoxon and Mann–Whitney tests for non-Gaussian variables), Spearman correlation test, and chi-square test were employed. Analyses were performed using the SPSS 28.0 software package (Chicago, IL, USA). All reported *p* values are two-sided, with *p* < 0.05 considered statistically significant.

### 2.5. Ethical Aspects

This study received approval from the Research Ethics Committee of the Fundación Española de Medicina Estética y Longevidad (FEMEL). The database was anonymized in accordance with the requirements set by Organic Law 3/2018 on Data Protection and Regulation (EU) 2016/679, ensuring that participants’ identities could not be directly or indirectly disclosed and that their data were handled securely. Additionally, the project complied with the ethical principles outlined in the Declaration of Helsinki, specifically those adopted at the 64th General Assembly in Fortaleza (Brazil), alongside prevailing national legislation on data analysis, protection, and the confidentiality of individuals.

## 3. Results

A total of 240 women participated in the analysis, with an average age of 45.1 ± 11.0 years (ranging from 20 to 70 years) and a mean BMI of 45.0 ± 6.5 kg/m^2^. Of these individuals, 135 participants (56.3%) underwent sleeve gastrectomy (SG), while 105 (43.7%) received a Roux-en-Y gastric bypass (RYGB). Regarding marital status, 156 women (65%) were married, 48 (20%) were single, and 36 (15%) were divorced. Twelve months after surgery, the mean BMI decreased to 29.5 ± 5.4 kg/m^2^, with an average excess BMI loss of 80.2 ± 19.4%. Table 1 outlines the anthropometric and sociodemographic variables by surgical technique, revealing that patients who underwent RYGB had a significantly lower mean BMI 24 months post-surgery and achieved a greater excess BMI loss. There were no significant differences in the distribution of comorbidities.

### 3.1. Results of FSFI Questionnaire

Participants exhibited significant improvements in the total FSFI score, with notable changes observed in the domains of arousal, lubrication, desire, orgasm, and satisfaction from baseline to 24 months post-surgery. However, no significant differences were identified in the pain domain (Table 2). Based on the FSFI-defined criterion for female sexual dysfunction (a total score of 26 or lower), 70.8% of participants were classified as having sexual dysfunction preoperatively, whereas only 20.5% retained this classification one year after surgery. Statistically significant enhancements were found across all domains except for pain.

There were no significant differences in the baseline or the postoperative values according to weight loss or the surgical technique they underwent.

### 3.2. Adherence to Mediterranean Diet—Impact on Weight Loss and Remission of Comorbidities

Through the use of the PREDIMED scoring system, 149 participants (62.1%) demonstrated poor adherence to the Mediterranean diet prior to surgery, while 91 (37.9%) exhibited medium adherence. Two years post-surgery, adherence levels improved significantly, with 165 patients (68.8%) achieving good adherence and 75 (31.2%) maintaining medium adherence (*p* < 0.001).

Patients with high adherence to the Mediterranean diet showed greater weight loss and a higher remission rate of type 2 diabetes mellitus, hypertension, and dyslipidemia, both after Roux-en-Y gastric bypass and sleeve gastrectomy (Table 3).

### 3.3. Association Between FSFI and Adherence to Mediterranean Diet

Before surgery, no significant associations were found between adherence to the Mediterranean diet and either the total FSFI score or any individual domain. However, at 24 months post-surgery, participants with good adherence to the Mediterranean diet displayed notably higher total FSFI scores (mean difference 2.1; 95%CI (1.7–2.5); *p* = 0.042). Domain-specific analysis revealed significant improvements in lubrication (mean difference 0.5; 95%CI (0.3–0.7); *p* = 0.012), orgasm (mean difference 0.5; 95%CI (0.2–0.8); *p* = 0.018), and satisfaction scores (mean difference 0.7; 95%CI (0.5–0.9); *p* = 0.006) among this group, whereas no substantial differences were observed in the remaining domains (Table 4).

Despite the fact that in all patients, significant improvement could be observed in all the domains after surgery, significant differences in the postoperative values could be detected in the domains of lubrication, orgasm, and satisfaction (Figure 2).

Multivariate analysis revealed that high adherence to the Mediterranean diet was independently associated with higher total FSFI scores (*p* = 0.048) and lubrication (*p* = 0.030), orgasm (*p* = 0.035), and desire (*p* = 0.018) scores.

### 3.4. Association Between FSFI and Presence of Postoperative Comorbidities

In regard to the association between the FSFI and the presence of postoperative comorbidities, the total FSFI score (mean difference 2.9; 95%CI (1.9–3.9); *p* = 0.044) and the domains of satisfaction (mean difference 1.0; 95%CI (0.5–1.5); *p* = 0.021) and lubrication (mean difference 0.9; 95%CI (0.5–1.4); *p* = 0.032) showed significantly higher values among patients without diabetes, including those without preoperative type 2 diabetes mellitus and those with complete remission (Table 5). There were no significant differences between subjects without previous type 2 diabetes mellitus and those with postoperative remission. Significant associations of FSFI values with the postoperative presence of comorbidities could not be demonstrated.

Multivariate analysis failed to demonstrate an independent association of the presence of postoperative type 2 diabetes mellitus with the total FSFI score or any of its domains.

## 4. Discussion

Historically, sexual dysfunction in individuals with obesity has received limited attention, largely due to its perception as a secondary issue and the discomfort associated with addressing such a personal topic. Nevertheless, the current literature highlights a high prevalence of sexual dysfunction in populations with obesity. Although this condition is more commonly reported in men, it is also significant among women, profoundly affecting their quality of life [27,28,29,30].

Several studies have demonstrated statistically significant improvements in sexual dysfunction scores following bariatric surgery, as measured by the FSFI questionnaire [31,32,33,34]. Aligning with our findings, Assimakopoulos et al. [31] observed notable enhancements across all FSFI domains at a 1-year follow-up, with a mean postoperative BMI of 31.8 kg/m^2^. Conversely, Sarwer et al. [35] reported that at 1 year post-surgery, significant improvements were limited to the desire and satisfaction domains, corresponding to a 32.7% reduction in weight. In our study, conducted 2 years post-surgery, participants achieved a mean postoperative BMI of 29.9 kg/m^2^, with an average excess BMI loss of 76.9% and a mean weight reduction of 34.1%, findings that are consistent with those reported by other studies at the 1-year mark.

Across multiple studies, the type of bariatric surgery performed did not appear to significantly influence improvements in sexual dysfunction scores, despite greater weight loss being observed with malabsorptive techniques [30]. This finding means that the amount of weight loss is not the main factor in the improvement in sexual dysfunction. Some studies hypothesize that self-reported sexual function begins to improve as early as 6 months postoperatively, coinciding with the period of optimal BMI change [36]. Bond et al. [32] reported that by 6 months post-surgery, most FSFI domains in patients following bariatric surgery were comparable to those of lean controls, with the exception of desire and lubrication. Conversely, Hernandez et al. [33] demonstrated that in a cohort of 80 patients undergoing biliopancreatic diversion, the mean FSFI score improved from 19.9 at baseline to 25.4 at 6 months, though this still indicated sexual dysfunction. However, by the 1-year follow-up, all participants had surpassed the threshold for sexual dysfunction, achieving a mean total FSFI score of 30.4. While sexual function improvements may be observed as early as 6 months after surgery, we believe that peak weight loss—which typically occurs between 12 and 24 months postoperatively—likely enhances these outcomes further. As a result, sexual dysfunction may continue to improve progressively during the 6 to 24 months following surgery. Although weight loss has not been shown to directly predict improvements in sexual dysfunction, it is plausible that better fitness associated with lower weight indirectly contributes to better sexual function, as does the adoption of healthy lifestyle habits and proper stress management. Future research should explore sexual function improvements at various intervals throughout the postoperative period to gain deeper insights. Our findings align with the meta-analysis conducted by Gao et al. [15], which concluded that neither baseline FSFI scores nor the degree of weight loss was a significant predictor of changes in total FSFI scores following bariatric surgery.

Olivera et al. [37] and Ranasinghe et al. [38] were the only studies that reported no improvement in sexual dysfunction following bariatric surgery. This lack of improvement was attributed to the inclusion of patients with persistent pelvic floor dysfunction, despite postoperative weight loss. The female pelvic floor plays a critical role in sexual function, and any damage to this area can result in the denervation of erectile tissue, ultimately leading to sexual dysfunction. For patients without pelvic floor disorders, various predictors of improvement in sexual dysfunction have been investigated. The Assimakopoulos research group found that better pre-existing sexual function and a lower pre-surgical BMI were associated with improved postoperative sexual function [31]. Conversely, Sarwer et al. [35] observed that patients with the poorest preoperative sexual function showed the most substantial postoperative improvements. In our study, no significant relationship was identified between preoperative and postoperative sexual function outcomes.

To the best of our knowledge, this study is the first to investigate how postoperative adherence to the Mediterranean diet improves female sexual function. In the general population, numerous studies have established a connection between higher adherence to the Mediterranean diet and improved female sexual health, along with a reduced prevalence of sexual dysfunction. Adopting a healthy dietary pattern, such as the Mediterranean diet, may have a positive influence on female sexual well-being [39,40].

Focusing on the FSFI questionnaire as a tool for assessing sexual function, Esposito et al. conducted a prospective interventional study on patients with metabolic syndrome to evaluate the effects of adopting a Mediterranean diet. In their intervention group, the mean FSFI score increased significantly from 19.7 at baseline to 26.1 post-treatment. However, this improvement was only observed in the overall FSFI score and not in any of the individual domains [41]. Conversely, other studies have demonstrated that higher adherence to the Mediterranean diet is linked to broader enhancements in female sexual health. Women following this dietary pattern reported better lubrication and sexual satisfaction, along with reduced pain during intercourse. Our results agree with this affirmation, as patients with good adherence to the Mediterranean diet show better lubrication and sexual satisfaction. In contrast, there were no significant differences in the pain domain. However, pain was the only domain that did not improve after surgery either. The Mediterranean diet has been linked to improved sexual function in women due to its anti-inflammatory, antioxidant, and vasodilatory properties. This diet can enhance blood flow and reduce inflammation, leading to better lubrication and sexual satisfaction. Additionally, the diet’s emphasis on antioxidants might modulate the release of anti-inflammatory agents involved in painful sensations during sexual intercourse. In any case, pain is subjective perception, and psychological factors can have a high influence on it [39]. Giugliano et al. assessed sexual function in women with type 2 diabetes and found a significant association between strong adherence to the Mediterranean diet and higher overall FSFI scores. Specifically, the satisfaction domain showed significantly higher values, with trends toward improvement observed in the lubrication and orgasm domains [25]. Similarly, our study demonstrated that good postoperative adherence to the Mediterranean diet was associated with higher total FSFI scores, as well as significant improvements in the lubrication, orgasm, and satisfaction domains.

The precise mechanisms through which a Mediterranean-style diet enhances sexual function in women are not yet fully understood. It is known that macronutrient intake can trigger oxidative stress, contributing to a pro-inflammatory state. Additionally, dietary fiber—a key component of the Mediterranean diet—may influence cytokine levels, helping to reduce inflammation. Dietary fiber is recognized for its anti-inflammatory properties, and its combination with antioxidant-rich foods in the Mediterranean diet likely mitigates oxidative stress associated with macronutrient consumption. From a public health perspective, understanding the exact mechanisms of individual nutrients or foods may be less critical than emphasizing holistic dietary patterns. Current disease prevention guidelines advocate for comprehensive dietary changes, such as reducing fat intake while increasing the consumption of whole grains and vegetables. These recommendations underscore the potential for synergistic effects of various dietary components in promoting health [42,43,44].

### Limitations

One inherent limitation of self-reported assessments of intimate behavior is the potential for response bias. Participants may feel uncomfortable or embarrassed while completing sexual functioning questionnaires, or they may misinterpret certain items. Consequently, addressing sexual dysfunction should not rely solely on a single measurement tool; incorporating multiple measures enhances the robustness of evidence. It is recommended to supplement the data from these questionnaires with additional tests that evaluate other facets of personal life or assess overall quality of life. Cultural or psychological factors that might influence sexual function were not assessed in this study. Psychosocial improvements after surgery, such as enhanced self-esteem, might mediate FSFI gains, for example. Future studies must complement the information obtained by means of the FSFI questionnaire with psychosocial aspects.

Moreover, these tools capture the level of sexual function only at the time of assessment and do not provide a comprehensive or long-term perspective on functionality. Furthermore, adherence to the Mediterranean diet was evaluated using a questionnaire. However, there is a possibility that patients may overestimate their adherence to the diet, potentially to portray themselves as compliant with prescribed dietary recommendations to healthcare professionals.

Finally, a better sexual function profile was observed in those patients without postoperative diabetes. Although good adherence to the Mediterranean diet appears as an independent factor for improved sexual function, adherence to the diet also increases the remission rate of T2DM. Adherence to the Mediterranean diet may need to be considered as part of a holistic lifestyle intervention that achieves improvements in multiple aspects of both physical and psychological health.

## 5. Conclusions

After bariatric surgery, the overall FSFI score increased significantly, as well as scores for each of its domains, except for pain, which was not altered. The improvement in sexual function is not influenced by weight loss, nor by the surgical technique used.

Postoperative good adherence to the Mediterranean diet is associated with better global FSFI scores, with significant improvements in the domains of lubrication, orgasm, and satisfaction. Enhancing sexual function in women can be included among the potential benefits linked to following a traditional Mediterranean-style diet, even in patients after bariatric surgery. Future studies must investigate additional variables such as psychological factors, physical activity, and other lifestyle changes that may also influence sexual function.

## Figures and Tables

**Figure 1 nutrients-17-00605-f001:**
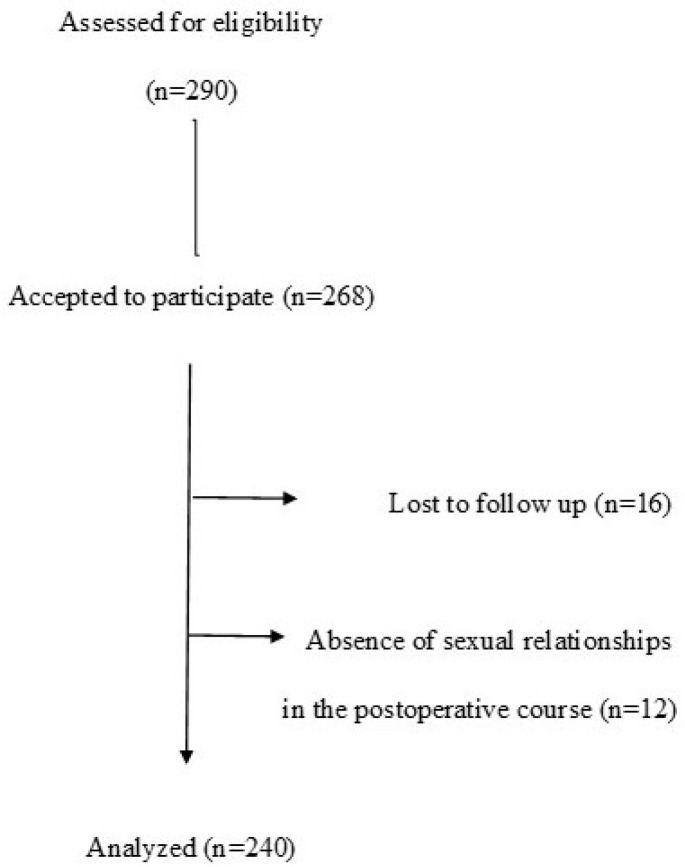
Flow diagram.

**Figure 2 nutrients-17-00605-f002:**
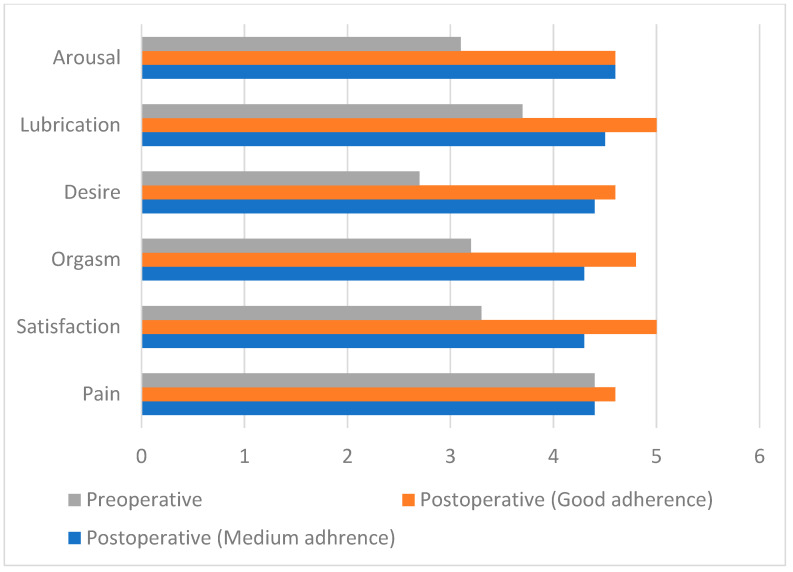
Pre- and postoperative differences in FSFI domains across adherence groups.

**Table 1 nutrients-17-00605-t001:** Distribution of anthropometric and sociodemographic variables and preoperative comorbidities depending on surgical technique.

	SG (n = 135)	RYGB (n = 105)	*p*
Age (years)	45.7 ± 10.6	43.9 ± 11.0	0.083
Marital status (N-%)			0.703
- Single	27 (20%)	21 (20%)	
- Married	90 (66.7%)	66 (62.9%)	
- Divorced	18 (13.3%)	18 (17.1%)	
Baseline BMI (kg/m^2^)	44.3 ± 6.0	45.5 ± 6.8	0.092
24-month BMI (kg/m^2^)	30.7 ± 6.6	28.4 ± 3.8	0.002
24-month excess BMI loss (%)	76.4 ± 21.2	83.5 ± 17.3	0.005
Type 2 diabetes mellitus (%)	39 (28.9%)	34 (32.4%)	0.559
Arterial hypertension (%)	53 (39.3%)	48 (45.7%)	0.315
Dyslipidemia (%)	48 (35.6%)	45 (42.9%)	0.249

SG: sleeve gastrectomy; RYGB: Roux-en-Y gastric bypass.

**Table 2 nutrients-17-00605-t002:** Changes in FSFI scores.

	Baseline	24 Months Postoperatively	*p*
Total FSFI score	20.4 ± 10.8	27.9 ±11.1	<0.001
Arousal	3.1 ± 1.6	4.6 ± 1.7	<0.001
Lubrication	3.7 ± 1.7	4.8 ± 1.9	<0.001
Desire	2.7 ± 1.3	4.5 ± 1.4	<0.001
Orgasm	3.2 ± 2.0	4.6 ± 2.0	<0.001
Satisfaction	3.3 ± 1.9	4.8 ± 2.0	<0.001
Pain	4.4 ± 2.5	4.6 ± 2.3	0.230

**Table 3 nutrients-17-00605-t003:** Weight loss and remission of comorbidities depending on adherence to Mediterranean diet.

	Sleeve Gastrectomy (n = 135)			Roux-en-Y Gastric Bypass (n = 105)		
	Good Adherence (n = 94)	Medium Adherence (n = 41)	*p*	Good Adherence (n = 71)	Medium Adherence (n = 34)	*p*
24-month excess BMI loss (%)	80.6 ± 21.4	72.1 ± 20.1	0.034	87.1 ± 19.0	80.2 ± 17.1	0.038
Type 2 diabetes mellitus remission	92.8% 26 out of 28	63.6% 7 out of 11	0.023	95.8% 23 out of 24	70% 7 out of 10	0.033
Arterial hypertension remission	78.4% 30 out of 37	50% 8 out of 16	0.021	88.5% 31 out of 35	61.5% 8 out of 13	0.032
Dyslipidemia remission	85.3% 29 out of 34	57.1% 8 out of 14	0.035	96.9% 31 out of 32	76.9% 10 out of 13	0.035

**Table 4 nutrients-17-00605-t004:** Associations between the different domains of the FSFI and adherence to the Mediterranean diet.

	Good Adherence (n = 165)	Medium Adherence (n = 75)	*p*
Total FSFI score	28.6 ± 11.3	26.5 ± 11.0	0.042
Arousal	4.6 ± 1.6	4.6 ± 1.7	0.568
Lubrication	5.0 ± 1.9	4.5 ± 1.8	0.012
Desire	4.6 ± 1.4	4.4 ± 1.3	0.257
Orgasm	4.8 ± 2.0	4.3 ± 1.9	0.018
Satisfaction	5.0 ± 2.0	4.3 ± 2.0	0.006
Pain	4.6 ± 2.1	4.4 ± 2.3	0.281

**Table 5 nutrients-17-00605-t005:** Associations between the different domains of the FSFI and the presence of postoperative Type 2 diabetes mellitus.

	Non-Diabetic (n = 230)	Diabetic (n = 10)	*p*
Total FSFI score	28.7 ± 11.1	25.8 ± 15.0	0.044
Arousal	4.7 ± 1.6	4.6 ± 2.1	0.683
Lubrication	5.0 ± 1.8	4.1 ± 2.0	0.032
Desire	4.6 ± 1.3	4.4 ± 1.8	0.579
Orgasm	4.8 ± 2.0	4.3 ± 2.3	0.091
Satisfaction	5.0 ± 1.9	4.0 ± 2.3	0.021
Pain	4.6 ± 2.0	4.4 ± 2.4	0.683

## Data Availability

Data are unavailable due to privacy or ethical restrictions.

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
