# Peer review of "Changes in Sexual Functioning in Women with Severe Obesity After Bariatric Surgery: Impact of Postoperative Adherence to Mediterranean Diet"

_nutrients, 2025, doi:10.3390/nu17040605_

Round 1
Reviewer 1 Report
Comments and Suggestions for Authors
Thanks for the article, I appreciate very much the work on a subject unfortunately considered very marginal in medical research. However, you will have to make some improvements. First, about the diet: the inclusion of Mediterranean diet adherence feels underdeveloped and insufficiently valorized. You might explore more mechanisms linking Mediterranean diet adherence to specific FSF domains (e.g., lubrication, orgasm). Maybe provide more comparative data, such as differences in weight loss, psychological health, or comorbidities between high and medium adherence groups, to contextualize the diet's impact on FSF. Some other factors to discuss are confounders like socioeconomic factors or individual motivation that might affect adherence and outcomes. Now, regarding statistics, they are basic, lacking depth and visual representation. Employ more advanced statistical methods, such as multivariate regression or mediation analysis, to determine if adherence to the Mediterranean diet independently predicts FSF improvements or if it acts through other variables (e.g., weight loss, psychological well-being). Present effect sizes and confidence intervals to provide more clarity on the magnitude of observed changes. Incorporate interaction effects (e.g., adherence to diet × type of surgery) to explore nuanced relationships. The manuscript also lacks engaging and informative visuals to complement the data. Please, include graphics, comparative plots (e.g., bar charts, lines) to highlight pre- and post-surgery differences in FSF scores across adherence groups.
Also, while some limitations are acknowledged, others are overlooked. Acknowledge cultural or psychological factors influencing sexual function that were not assessed. Discussions can be more elaborate, in order to improve the value of your research. Investigate whether psychosocial improvements post-surgery, such as enhanced self-esteem, might mediate FSF gains, for example. Or/and reflect on whether the Mediterranean diet could be part of a holistic lifestyle intervention post-surgery.
Author Response
First, about the diet: the inclusion of Mediterranean diet adherence feels underdeveloped and insufficiently valorized. You might explore more mechanisms linking Mediterranean diet adherence to specific FSF domains (e.g., lubrication, orgasm). Maybe provide more comparative data, such as differences in weight loss, psychological health, or comorbidities between high and medium adherence groups, to contextualize the diet's impact on FSF.
ANSWER: It has been specified that “Domain-specific analysis revealed significant improvements in lubrication, orgasm, and satisfaction scores among patients with high adherence to Mediteranean diet, whereas no substantial differences were observed in the remaining domains” (Section 3.3, 1st paragraph and Table 4).
The distribution of comorbidities (Table 1) and the remission rate (Table 3) have been added. A significant association between FSFI and the postoperative presence of T2DM has been also established (Section 3.4).
Some other factors to discuss are confounders like socioeconomic factors or individual motivation that might affect adherence and outcomes. Now, regarding statistics, they are basic, lacking depth and visual representation. Employ more advanced statistical methods, such as multivariate regression or mediation analysis, to determine if adherence to the Mediterranean diet independently predicts FSF improvements or if it acts through other variables (e.g., weight loss, psychological well-being). Present effect sizes and confidence intervals to provide more clarity on the magnitude of observed changes. Incorporate interaction effects (e.g., adherence to diet × type of surgery) to explore nuanced relationships. The manuscript also lacks engaging and informative visuals to complement the data. Please, include graphics, comparative plots (e.g., bar charts, lines) to highlight pre- and post-surgery differences in FSF scores across adherence groups.
ANSWER: Multivariate analysis was performed for the association of FSFI score and the subdomains with adherence to the Mediterranean diet and the presence of postoperative T2DM. “Multivariate analysis revealed that the high adherence to the Mediterranean was independently associated with higher levels of total FSFI score (p=0.048), lubrication (p=0.030), orgasm (p=0.035) and desire (p=0.018).” (Section 3.3, 2nd paragraph). “Multivariate analysis failed to demonstrate an independent association of the presence of postoperative Type 2 diabetes mellitus with the total FSFI score or any of its domains.” (Section 3.3, 2nd paragraph).
Confidence intervals have been added for the associations between FSFI and adherence to the Mediterranean diet (Section 3.3, 1st paragraph) and the presence of T2DM (Section 3.4, 1st paragraph).
A graphic highlighting pre- and post-surgery differences in FSFI domains across adherence groups has been added (Figure 1).
Also, while some limitations are acknowledged, others are overlooked. Acknowledge cultural or psychological factors influencing sexual function that were not assessed. Investigate whether psychosocial improvements post-surgery, such as enhanced self-esteem, might mediate FSF gains, for example. Or/and reflect on whether the Mediterranean diet could be part of a holistic lifestyle intervention post-surgery.
ANSWER: We agree with the comment of the reviewer. Unfortunately, cultural or psychological factors were not recorded in the study. This has been added as a limitation “Cultural or psychological factors that might influence sexual function were not assessed in the study. Psychosocial improvements after surgery, such as enhanced self-esteem, might mediate FSFI gains, for example. Future studies must complete the information obtained by means of the FSFI questionnaire with psychosocial aspects.”(Limitations section 1st paragraph).
The following paragraph has been added “Finally, a better sexual function profile was observed in those patients without postoperative diabetes. Although good adherence to the Mediterranean diet appears as an independent factor for improved sexual function, adherence to the diet also increases the remission rate of T2DM. Adherence to the Mediterranean diet may need to be considered as part of a holistic lifestyle intervention that achieves improvements in multiple aspects of both physical and psychological health.” (Limitations section 3rd paragraph).
Reviewer 2 Report
Comments and Suggestions for Authors
In the current article the authors studied the effect of bariatric surgery on female sexual functioning evaluated by FSFI and determined the impact of the postoperative adhesion to a mediterranean diet. They concluded that after bariatric surgery, the overall FSFI score increased significantly and that the greatest improvements are observed among women with postoperative good adherence to mediterranean diet.
Some suggestions:
1. Add please as Supplementary material the 2 questionnaires (FSFI and PREDIMED).
2. Lines 103-104: please clarify what do you mean by “Those studies lost to follow-up were removed from the statistical analysis”.
3. Lines 141-143 – please give more details concerning the “diet to be followed”.
4. Did the patient have problems with the diet? What kind? Please specify.
5. Point 2.5 Ethical aspects: add please the No of the ethical approval and the date in which the study was approved .
The authors underlined that there are no studies reported in the literature which evaluates the improvement of sexual function after bariatric surgery depending on the adherence to the Mediterranean diet. Therefore the study is interesting and welcome.
Author Response
In the current article the authors studied the effect of bariatric surgery on female sexual functioning evaluated by FSFI and determined the impact of the postoperative adhesion to a mediterranean diet. They concluded that after bariatric surgery, the overall FSFI score increased significantly and that the greatest improvements are observed among women with postoperative good adherence to mediterranean diet.
Some suggestions:
- Add please as Supplementary material the 2 questionnaires (FSFI and PREDIMED).
ANSWER: The two questionnaires have been added as Supplementary material.
- Lines 103-104: please clarify what do you mean by “Those studies lost to follow-up were removed from the statistical analysis”.
ANSWER: The sentence has been corrected to “Cases lost to follow-up were omitted from the analysis.” (Material and methods, 1st paragraph).
- Lines 141-143 – please give more details concerning the “diet to be followed”.
ANSWER: More details concerning the diet have been added “The solid diet adhered to the Mediterranean diet framework, with the following characteristics: using olive oil as the main source of fat, consuming plenty of fruit, vegetables, pulses and nuts, including wholemeal bread, pasta, rice and other wholegrain cereals in the daily diet, opting for low-processed, fresh and locally sourced foods, consuming dairy products daily, such as low-fat yoghurt and fresh cheese, consume red meat in moderation, preferably as part of stews, eat plenty of fish and eggs in moderation, fresh fruit should be the usual dessert, and water should be the only beverage consumed, although water should be drunk between meals, not during. Eat slowly, chewing food well, and divide meals into several small meals a day. ” (Section 2.2, 2nd paragraph)
- Did the patient have problems with the diet? What kind? Please specify.
ANSWER: During the first year after surgery, patients usually have difficulties in eating red meat, rice and pasta, which usually disappear 12 months after surgery, and they can usually eat all types of food without problems. (Section 2.2, 3rd paragraph).
- Point 2.5 Ethical aspects: add please the No of the ethical approval and the date in which the study was approved .
ANSWER: “(protocol code 2020/3; date of approval June 10th, 2020).” (Institutional review board statement, After Conclusion section)

Reviewer 3 Report
Comments and Suggestions for Authors
All comments, suggestions, and questions are available throughout the manuscript.

Author Response
- FSFI has been written in extensive form, presenting the abbreviation
ANSWER: It has been corrected (Abstract)
- Provide challenge for the next study
ANSWER: This sentence has been added “Future studies must investigate additional variables such as psychological factors, physical activity, and other lifestyle changes that may also influence sexual function.” (Abstract and Conclusion, 2nd paragraph)
- Keywords: Severe obesity; Bariatric surgery; Female Sexual Function Index; Mediterranean diet; Adherence; Lubrication; Orgasm; Satisfaction.
ANSWER: Type 2 diabetes mellitus and Metabolic syndrome have been added.
- ANSWER: It has been added “This retrospective observational study utilized a database compiled prospectively. The cohort included heterosexual Spanish women who underwent Roux-en-Y gastric bypass (RYGB) or sleeve gastrectomy (SG) between September 2020 and September 2022 at a Spanish private institution” (Materials and methods, 1st paragraph)
- ANSWER: Section 3.1: The format has been changed to normal.
- ANSWER: We prefer to maintain Tables 2 and 4 separately, as we think they are easier to understand.
- ANSWER: Table 4 (previously Table 3) had a wrong row. This has been removed.
- ANSWER: The sentence “This finding implies that weight loss alone may not be the primary factor driving improvements in sexual dysfunction.” has been replaced by “this finding means that the amount of weight loss is not the main factor in the improvement of sexual dysfunction.” (Discussion, 3rd paragraph).
- Discussion: Other factors associated with sexual improvement
ANSWER: Although weight loss has not been shown to directly predict improvements in sexual dysfunction, it is plausible that better fitness associated with lower weight indirectly contributes to better sexual function, as does the adoption of healthy lifestyle habits and proper stress management. (Discussion, 3rd paragraph)
- ANSWER: This sentence has been modified according to the reviewer´s suggestion “To the best of our knowledge, this study is the first to investigate how postoperative adherence to the Mediterranean diet improves female sexual function.” (Discussion, 5th paragraph).
- Women following Mediterranean diet reported better lubrication and sexual satisfaction, along with reduced pain during intercourse. Discuss the results presented in Table 2.
ANSWER: The following sentences have been added “Our results agree with this affirmation, as patients with good adherence to the Mediterranean diet show better lubrication and sexual satisfaction. In contrast, there were no significant differences in the pain domain. However, pain was the only domain that did not improve after surgery either. The Mediterranean diet has been linked to improved sexual function in women due to its anti-inflammatory, antioxidant, and vasodilatory properties. This diet can enhance blood flow and reduce inflammation, leading to better lubrication and sexual satisfaction. Additionally, the diet's emphasis on antioxidants might modulate the release of anti-inflammatory agents involved in painful sensations during sexual intercourse. Anyway, pain is subjective perception and psychological factors can have a high influence on it” (Discussion, 6th paragraph).